# Perceive, Reflect and Understand Long Video: Progressive Multi-Granular Clue Exploration with Interactive Agents

## Abstract

Long videos, characterized by temporal complexity and sparse task-relevant information, pose significant reasoning challenges for AI systems. Although various Large Language Model (LLM)-based approaches have advanced long video understanding, they still struggle to achieve both completeness and efficiency in capturing task-critical information. Inspired by human progressive visual cognition, we propose CogniGPT, a framework that leverages an interactive loop between Multi-Granular Perception Agent (MGPA) and Verification-Enhanced Reflection Agent (VERA) for efficient and reliable long video understanding. Specifically, MGPA mimics human visual divergent and focused attention to capture task-related information, while VERA verifies perceived key clues to mitigate hallucination and optimize subsequent perception strategies. Through this interactive process, CogniGPT explores a minimal set of informative and reliable task-related clues. Extensive experiments on EgoSchema, Video-MME, NExT-QA, and MovieChat datasets demonstrate CogniGPT's superiority in both accuracy and efficiency. Notably, on EgoSchema, it surpasses existing training-free methods using only 11.2 frames and achieves performance comparable to Gemini 1.5-Pro.

## 1 Introduction

Benefiting from the intricate mechanisms of brain and cognition, human intelligence fundamentally excels at effortlessly comprehending complex multimodal content (including natural language and hours-long videos) while performing sophisticated reasoning. However, this capability poses significant challenges for artificial intelligence systems, particularly in understanding and reasoning about dynamic visual content alongside textual information. With advancements in computer vision, various multimodal video tasks have been extensively studied, such as video question answering (Bai et al., 2023; Kim et al., 2023), moment retrieval (Moon et al., 2023; Xu et al., 2024), and video captioning (Zhao et al., 2023). Nevertheless, these tasks typically focus on isolated aspects of video analysis and struggle to achieve comprehensive multimodal understanding and reasoning, remaining far from realizing Artificial General Intelligence.

With the rapid advancement of Large Language Models (LLMs), harnessing their commonsense knowledge and reasoning capabilities for complex video understanding has garnered increasing attention. Existing approaches can be broadly categorized into two paradigms: Multimodal Large Language Model (MLLM)-based methods and LLM Agent-based methods. MLLM-based methods (Li et al., 2023; Zhang et al., 2023b; Maaz et al., 2023) align visual and textual tokens through end-to-end training, enabling LLMs to comprehend video content. However, these methods encounter substantial challenges when processing long videos, primarily due to the difficulty of effectively balancing spatial-temporal details and capturing long-range dependencies within a limited number of visual tokens. Although some MLLM-based methods (Song et al., 2024; He et al., 2024) adopt various compression strategies to partially alleviate these issues, they still suffer from hallucinations (Ma et al., 2024a), incur high training costs, and lack interpretable reasoning capabilities for complex tasks.

More recently, several pioneering works attempt to use LLMs as agents for video understanding (Fan et al., 2024; Zhang et al., 2023a; Ma et al., 2024b), demonstrating significantly enhanced reasoning

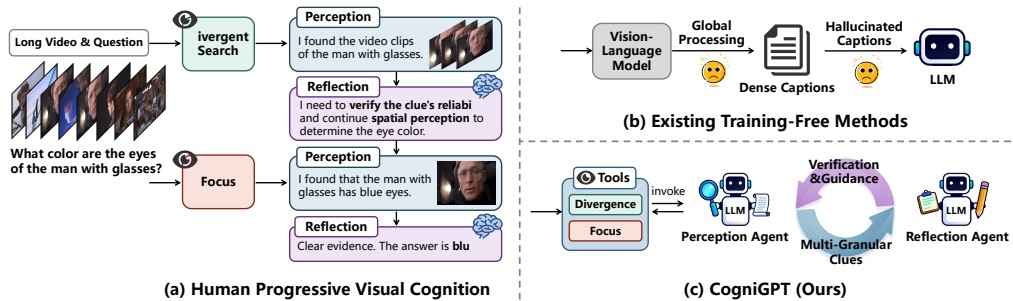

Figure 1: (a) Humans comprehend long videos through iterative interactions between *perception* and *reflection*, leveraging only a small amount of key information. (b) Existing training-free methods inefficiently perform global captioning and are susceptible to caption hallucinations from vision-language models. (c) CogniGPT (Ours) mimics human cognition by employing interactive LLM agents to extract reliable and minimal key information.

capabilities in a training-free manner. These methods typically employ pre-trained video-language models to preprocess the entire video into structured memory representations (e.g., dense captions), and then apply LLMs to perform reasoning tasks. However, such global processing is inefficient for long videos and introduces substantial redundant information, which hinders the reasoning capability of LLMs. Although VideoAgent* (Wang et al., 2024) and VideoTree (Wang et al., 2025b) avoid global video processing, they rely solely on single-granularity retrieval and captions, which depend on retrieval and caption performance and are susceptible to caption hallucinations. Thus, existing methods fail to achieve an optimal trade-off between information adequacy and computational efficiency.

To address the aforementioned challenges, we draw inspiration from the way humans perceive and understand videos. For humans, video comprehension is not about capturing every detail, but rather a progressive process of selecting key information. This process is guided by the *Dual Process Theory* (Kahneman, 2011). Specifically, the *Intuitive Thinking* system continuously guides perceptual behaviors and retains multi-granular, task-relevant clues in working memory (Baddeley, 1974). Subsequently, the *Analytical Thinking* system reflects on the perceived information, verifies the reliability of critical clues, and guides the subsequent round of perception. Through iterative interactions between perception and reflection, humans are able to progressively explore task-critical clues, enabling efficient and reliable reasoning.

Inspired by the aforementioned mechanisms, we propose CogniGPT, a novel framework that leverages LLM agents to emulate the human *Perception-Reflection* loop to progressively explore minimal yet comprehensive information for long video understanding. Specifically, we propose the Multi-Granular Perception Agent (MGPA), which iteratively invokes multi-granular perception actions to adaptively extract task-relevant information. Inspired by the focusing and diverging mechanisms of human vision, these actions—`temporal focus`, `spatial focus`, and `divergent search`—work in a complementary manner to capture multi-granular critical clues while filtering out irrelevant content. To mitigate the impact of hallucinations in vision–language models, we further design the Verification-Enhanced Reflection Agent (VERA), which leverages complementary multi-frame information with varying perspectives and granularities to validate critical clues, and provides verbal feedback to optimize the subsequent perception strategy. Through the interactive loop between MGPA and VERA, CogniGPT achieves accurate long-video understanding by analyzing only a minimal number of task-relevant frames. Extensive experiments on the EgoSchema (Mangalam et al., 2023), Video-MME (Fu et al., 2024), MovieChat (Song et al., 2024), and NExT-QA (Xiao et al., 2021) benchmarks demonstrate the superiority of CogniGPT in both accuracy and efficiency. Notably, on the EgoSchema benchmark, CogniGPT surpasses the accuracy of DrVideo (Ma et al., 2024b) by 2.8%, while using only 12.4% of the frames and 27.6% of the runtime. Furthermore, it performs competitively with the proprietary Gemini 1.5-Pro. In summary, the key contributions of this work are as follows:

- Inspired by human cognition, we construct a *Perception-Reflection* loop with LLM-based agents to progressively explore critical and reliable information for long video understanding.

- We design MGPA to adaptively perceive task-relevant information at multiple granularities, while VERA mitigates hallucination interference by cross-validating from multiple perspectives.

- Extensive experiments on EgoSchema, Video-MME, NExT-QA, and MovieChat datasets demonstrate the superiority of CogniGPT in both accuracy and efficiency.

## 2 RELATED WORK

### 2.1 MLLMS FOR LONG VIDEO UNDERSTANDING

Long video understanding is challenging due to spatio-temporal complexity and the sparsity of relevant information. The key challenge for MLLM-based methods lies in balancing spatial-temporal details and capturing long-range dependencies within a limited context window. As a result, these methods focus on token compression strategies (Hussein et al., 2019; Islam & Bertasius, 2022; Nguyen et al., 2022; Song et al., 2024; Wang et al., 2024). For example, MovieChat (Song et al., 2024) reduces token redundancy, and Chat-UniVi (Jin et al., 2024) applies kNN clustering for compression. Despite partially mitigating these issues, MLLM-based methods still suffer from hallucinations (Ma et al., 2024a), high training costs, and a lack of interpretable reasoning. In contrast, we employ LLMs as agents to emulate a human-like *Perception-Reflection* loop, progressively exploring reliable clues for long video understanding in a training-free manner.

### 2.2 LLM AGENTS FOR LONG VIDEO UNDERSTANDING

An agent is an entity that makes decisions and takes actions in a dynamic environment to achieve specific goals. Recent advances leverage LLM agents for video understanding, demonstrating improved reasoning capabilities without requiring additional training (Wang et al., 2023; Zhang et al., 2023a; Yang et al., 2024; Fan et al., 2024; Ma et al., 2024b). These approaches typically preprocess videos into structured memory representations using pre-trained video-language models (e.g., dense video captioning), followed by LLM-based reasoning. However, such global processing proves inefficient for long videos and introduces substantial redundancy, limiting LLM reasoning capabilities. While VideoAgent (Wang et al., 2024) and VideoTree (Wang et al., 2025b) avoid global captioning, they still rely on single-granularity retrieval and captions, making them dependent on retrieval and captioning performance, and susceptible to hallucinations. To address these limitations, we design MGPA to adaptively perceive task-relevant information at multiple granularities, alleviating reliance on single-granularity models. We also design VERA, which validates critical clues using complementary information from multiple frames and perspectives, mitigating hallucination interference in vision-language models.

## 3 METHODS

### 3.1 OVERVIEW

Given a long video $V$ and a complex query $Q$, humans do not process the entire video sequentially. Instead, they decompose $Q$ into sub-tasks and engage in a cognitive loop of *Perception* and *Reflection*, gradually identifying a minimal set of keyframes to answer $Q$. Inspired by this process, we propose CogniGPT, which simulates this cognitive loop through the interaction of the Multi-Granular Perception Agent (MGPA) and Verification-Enhanced Reflection Agent (VERA), as shown in Figure 2. Specifically, at each iteration $t$, MGPA $\pi_p$ selects an action $a_t$ (i.e., a tool from the *Multi-Granular Perception Toolkit*) and its corresponding input based on the query $Q$, the current *Working Memory* $\mathcal{M}_{t-1}$, and the guidance $g_t$ provided by VERA, following the policy $\pi_p(a_t \mid Q, \mathcal{M}_{t-1}, g_t)$. The selected action produces an observation $o_t$, which is incorporated into memory to update the state $\mathcal{M}_t$. VERA then verifies critical clues and provides verbal feedback to refine the subsequent perception strategy. We detail each component below.

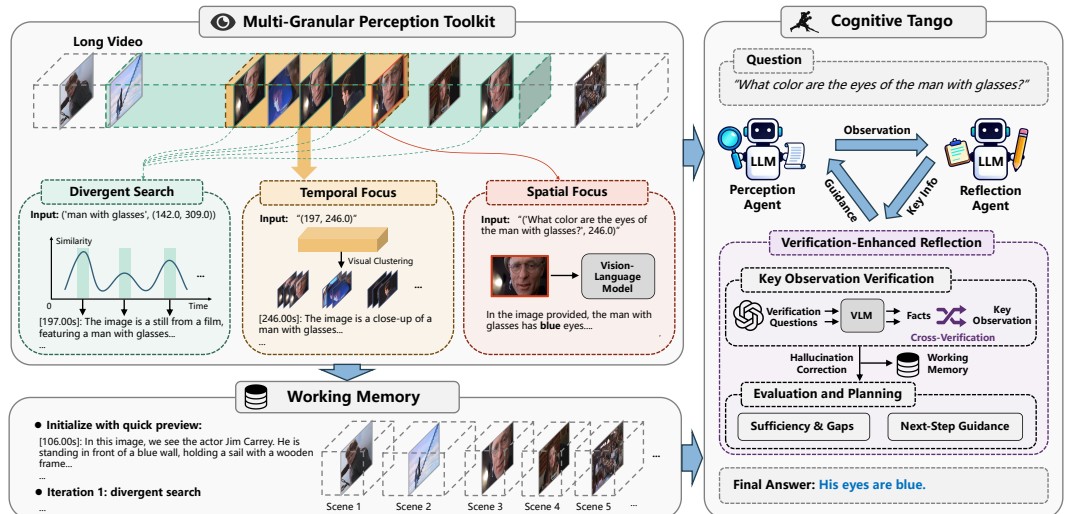

Figure 2: Overview of CogniGPT. **Left:** The *Multi-Granular Perception Toolkit* includes multimodal tools that simulate human visual mechanisms of focused and divergent attention. It extracts key information from both local and global perspectives, storing it as evidence in the *Working Memory*. **Right:** The *Cognitive Tango* progressively interprets long videos through iterative interaction between the Multi-Granular Perception Agent and the Verification-Enhanced Reflection Agent.

## 3.2 MULTI-GRANULAR PERCEPTION

Studies show that humans perceive the environment using visual attention mechanisms, enabling both global divergent search and focused attention on specific spatiotemporal regions to adaptively acquire task-relevant information (Corbetta & Shulman, 2002; Desimone et al., 1995). Inspired by this insight, we design an action space that enables the LLM to perceive task-relevant frames from multiple perspectives. The defined actions include `divergent search`, `temporal focus`, and `spatial focus`, described as follows:

**Divergent Search.** We simulate the divergent search mechanism in human visual attention through the `divergent search` tool, designed to identify key frames relevant to a subtask over a broad temporal range, avoiding redundant video captioning. Specifically, the LLM first infers a subtask query $q$ along with a target temporal span as the input to this tool. For example, if the main query $Q$ is "What color are the eyes of the man with glasses?", the LLM may extract "man with glasses" as the sub-query $q$.

We employ EVA-CLIP-8B (Sun et al., 2024) to extract the textual representation of $q$ and the visual representations of sampled video frames $v_i$ within the specified span. We then compute cosine similarity scores $s_i$ between $q$ and each $v_i$. To emulate the human divergent search, we select frames with broad contextual diversity, inspired by the watershed algorithm (Vincent & Soille, 1991). Specifically, we smooth the similarity scores $s_1, s_2, \ldots, s_T$ using a sliding-window average to obtain refined scores $\tilde{s}i = \text{smooth}(s_i)$. We then compute the mean similarity $\bar{s} = \frac{1}{T}\sum i = 1^T \tilde{s}_i$ as a threshold to segment the timeline into peak and valley regions. From each peak region, we select the frame with the highest $\tilde{s}_i$ score and retain the top-$N_f$ frames overall. Unlike existing methods (Fan et al., 2024) that use a top-k selection strategy and often pick redundant frames clustered around the highest peaks, our approach captures diverse, causally relevant contextual information, reducing retrieval redundancy.

After retrieving the $N_f$ key frames, we use a vision-language model to generate captions for each selected frame, along with their timestamps. This tool enables the LLM to efficiently locate and understand query-relevant visual evidence, including temporal aspects such as "when" certain events occur.

**Temporal Focus.** While `divergent search` effectively retrieves relevant video segments, its coarse temporal granularity can overlook crucial fine-grained details like rapid action transitions.

To address this, we introduce `temporal focus` as a complement, establishing a coarse-to-fine reasoning paradigm. After `divergent search` identifies a broad span, `temporal focus` performs a granular semantic analysis within it, capturing key sub-events and enabling a detailed understanding of dynamic content.

Specifically, to balance granularity and efficiency, we first employ EVA-CLIP-8B (Sun et al., 2024) to extract video features within the given temporal span. We then perform K-means clustering based on semantic similarity, obtaining $K_t$ clusters $C_1, \ldots, C_{K_t}$. For each cluster $C_i$, we select the clip nearest to its centroid $c_i$ as the representative and generate a caption using a vision–language model. This process yields $K_t$ captions that summarize the representative semantic content of the segment, thereby enabling fine-grained yet efficient temporal understanding.

**Spatial Focus.** Although we utilize `divergent search` and `temporal focus` to obtain the captions for frames or clips, the understanding of spatial content heavily relies on the quality of the captions. This reliance may result in the inability to capture fine-grained spatial details, attributes, and positional relationships. For instance, as shown in Figure 2, when the question is "What color are the eyes of the man with glasses?", the caption for the relevant frame does not include information about the color of the eyes, necessitating further spatial understanding. To address this issue, we propose the `spatial focus` tool, which enables fine-grained spatial understanding through Visual Question Answering (VQA). Specifically, we prompt the LLM to input the frame to be analyzed and a sub-task question, and leverage LLaVA-NeXT (Liu et al., 2024) to perform VQA on the frame, enabling the extraction of task-relevant spatial information that goes beyond what is captured in the initial caption.

It is worth emphasizing that although we perform global video feature extraction with EVA-CLIP-8B, along with similarity computation and clustering, the required runtime is negligible compared to caption generation and LLM inference. Supporting experimental evidence is presented in Section 4.5. The detailed prompt descriptions of the tools are provided in the Appendix.

### 3.3 Working Memory

Inspired by human working memory, we propose a dynamically updated *Working Memory* $\mathcal{M}$ to store perceived information, thus providing an evolving context for LLM-based planning and reasoning.

**Initialization with Quick Preview.** Before engaging in reasoning, humans typically form a general understanding of the video context. In a similar manner, we employ EVA-CLIP-8B (Sun et al., 2024) to extract video features, perform K-means clustering into $K_m$ clusters, and generate captions for the corresponding cluster centroids. These $K_m$ captions summarize the representative scenes within the video. The selected frames' timestamps and captions are integrated into $o_0$ to initialize $\mathcal{M}$, supplying essential contextual cues for LLM reasoning.

**Dynamic Update.** At each iteration $t$, the new action $a_t$ and observation $o_t$ are added to memory $\mathcal{M}$, forming $\mathcal{M}_t = (a_0, o_0, a_1, o_1, \ldots, a_t, o_t)$, which provides the LLM with progressively richer contextual cues.

### 3.4 Verification-Enhanced Reflection

Given that MGPA is susceptible to hallucinations from vision-language models (VLMs) and exhibits limited planning capabilities—often leading to suboptimal strategies—we propose the Verification-Enhanced Reflection Agent (VERA) to continuously detect and correct hallucinated perceptual information. Furthermore, it analyzes the global context and provides verbal feedback to optimize the action strategy of MGPA.

**Key Observation Verification.** Since reasoning depends on textual descriptions generated by the VLM, hallucinations in key observations may directly cause incorrect inference by the LLM. To address this issue, inspired by Chain-of-Verification (Dhuliawala et al., 2023), VERA performs cross-verification of critical observations. Specifically, at each iteration step $t$, VERA first determines whether the latest observation $o_{t-1}$ contains key information relevant to answering the question $Q$. If such information exists, the verification proceeds as follows: we prompt the LLM to freely generate a set of verification questions (2-3 in our experiments) from multiple perspectives based on the identified key information across frames. For example, if the original question $Q$ is "What color is the boy's hat in the video?" and a caption in a certain frame states "a boy is wearing a

red hat," then a possible verification question is "What color is the boy's hat in the image?" These verification questions together with their corresponding frames are input into the VLM for visual question answering, yielding multiple factual responses. The LLM is then prompted to perform cross-verification between these factual responses and the original key information to assess its reliability. If the information is deemed trustworthy, it is stored in the *Working Memory*. This verification mechanism essentially leverages complementary multi-frame information—capturing different viewpoints and granularities—to eliminate hallucinatory content.

**Evaluation and Planning.** Subsequently, VERA performs a chain-of-thought (CoT) analysis over the question $Q$ and the current *Working Memory* $\mathcal{M}_{t-1}$, evaluating key aspects, including: *Information Sufficiency*, which examines whether the available information is sufficient to reliably answer the question; *Information Gaps*, identifying any critical missing details; and *Next Step Decision*, determining whether the agent should proceed with further information gathering or conclude with a final answer. If the decision is to continue, VERA generates a guidance output $g_t$, which specifies the next piece of information to collect and suggests the most effective tools for doing so. This guidance is then provided as verbal feedback to MGPA. If the decision is to terminate, VERA offers a final explanation and provides the answer. Additionally, a maximum iteration limit $T_{\max}$ is set, upon which a final answer is produced based on the currently available information.

# 4 EXPERIMENTS

## 4.1 DATASETS AND METRICS

We evaluate our approach on four established benchmarks: EgoSchema (Mangalam et al., 2023), with 500 samples from the official subset averaging 3 minutes; Video-MME (Fu et al., 2024), using the long split of the 300-video open-domain subset averaging 41 minutes; NExT-QA (Xiao et al., 2021), with 600 instances across causal, temporal, and descriptive types averaging 44 seconds; and MovieChat (Song et al., 2024), using the global mode of the official test split with 170 samples averaging 10 minutes. These datasets span diverse durations, question types, and reasoning tasks, enabling a comprehensive evaluation of CogniGPT. Full details are provided in the Appendix.

For EgoSchema, NExT-QA, and Video-MME, we evaluate accuracy on multiple-choice questions. For MovieChat, we use GPT-assisted accuracy evaluation (true/false). Following DrVideo (Ma et al., 2024b), we select Gemini-Pro (Team et al., 2023) as the evaluation assistant and adopt the same prompt (Maaz et al., 2023) for a fair comparison. Considering that captioning and QA are the primary sources of runtime, to compare the efficiency of training-free LLM Agent approaches, we also report the average number of frames requiring captioning and QA per sample.

## 4.2 IMPLEMENTATION DETAILS

To balance efficiency and accuracy, we configure CogniGPT with the following settings: On EgoSchema, NextQA, and MovieChat, we set $N_f = 5$, $K_t = 3$, and $K_m = 5$; on VideoMME, we set $N_f = 8$, $K_t = 5$, and $K_m = 5$. The maximum number of interaction iterations is set to $T_{\max} = 3$. We preprocess the raw videos at 1 FPS for the EgoSchema and NExT-QA benchmarks, 0.5 FPS for MovieChat, and 0.125 FPS for Video-MME. All experiments are conducted on 4 (or fewer) NVIDIA A6000 GPUs. Error analysis is provided in the Appendix.

## 4.3 MAIN RESULTS

### 4.3.1 COMPARISON WITH TRAINING-FREE LLM AGENTS

We perform a comprehensive comparison between our proposed method and existing training-free LLM agents, as summarized in Table 1. To ensure a fair comparison, we adopt the same vision-language models across all methods: we use LaViLa (Zhao et al., 2023) on EgoSchema to generate captions for 1-second video clips, CogAgent (Hong et al., 2024) on NExT-QA, and LLaVA-NeXT (Liu et al., 2024) on both Video-MME and MovieChat. The results demonstrate that our method consistently outperforms existing approaches in terms of both **efficiency** and **accuracy**.

Specifically, compared to global processing approaches (e.g., dense captioning), including LLoVi (Zhang et al., 2023a), VideoAgent (Fan et al., 2024), and DrVideo (Ma et al., 2024b),

Table 1: Results on the EgoSchema, Video-MME long split, NExT-QA, and MovieChat benchmarks. Accuracy is reported in %, and "Frame" denotes the average number of frames per sample that require captioning and QA.

| Method | LLM | EgoSchema | | Video-MME | | NExT-QA | | MovieChat | |
|---|---|---|---|---|---|---|---|---|---|
| | | Acc. | Frames | Acc. | Frames | Acc. | Frames | Acc. | Frames |
| **Training-Free LLM Agent** | | | | | | | | | |
| LLoVi (Zhang et al., 2023a) | GPT-3.5 | 51.8 | 180 | 45.4 | 492 | 67.7 | 22 | 58.3 | 180 |
| VideoAgent* (Wang et al., 2024) | GPT-4 | 60.2 | **8.4** | 46.4 | 24.6 | 71.3 | **8.2** | 58.1 | - |
| VideoAgent (Fan et al., 2024) | GPT-4 | 62.8 | >90 | - | - | 70.8 | 22 | - | - |
| VideoTree (Wang et al., 2025b) | GPT-4 | 66.2 | 62.4 | 53.1 | 128.0 | 73.5 | 12.6 | - | - |
| DrVideo (Ma et al., 2024b) | GPT-4 | 66.4 | >90 | 51.7 | >492 | - | - | 93.1 | > 492 |
| Ours | GPT-4 | **69.2** | 11.2 | **54.7** | **18.3** | **76.8** | 10.9 | **95.3** | **10.5** |

our method's advantages primarily lie in its efficiency. For example, on Video-MME, LLoVi and DrVideo require captions for at least 492 frames, while our method achieves better accuracy with an average of only 18.3 frames, significantly reducing the number of frames that need to be processed. This efficiency is attributed to our *Perception-Reflection* mechanism, which selectively identifies key frames, thus avoiding redundant global frame processing and ensuring comprehensive coverage of task-relevant information for accurate question answering. Additionally, our method also surpasses global processing approaches in terms of accuracy. On EgoSchema, our method achieves 6.4% and 2.8% higher accuracy than VideoAgent and DrVideo, respectively. This improvement is mainly due to our method's ability to eliminate redundant information and caption hallucinations, which otherwise interfere with LLM reasoning.

In comparison to the similarity-based key frame selection approach used by VideoAgent* (Wang et al., 2024), our method requires only 18.3 frames on the long video dataset Video-MME, demonstrating superior efficiency. This is because our method more accurately captures frames causally relevant to the task. While our method uses slightly more frames on shorter-duration datasets like EgoSchema and NExT-QA, it achieves 9.0% and 5.5% higher accuracy, respectively. This is primarily because our method extracts spatiotemporal information at multiple granularities and eliminates hallucinations in captions. Compared to the clustering-based key frame selection method in VideoTree (Wang et al., 2025b), our method requires signifi-

Table 2: Comparison with Proprietary and Open-Source MLLMs on EgoSchema and Video-MME long split.

| Method | EgoSchema | | Video-MME | |
|---|---|---|---|---|
| | Acc. | Frames | Acc. | Frames |
| **Proprietary MLLM** | | | | |
| Gemini 1.5-Flash | 65.7 | 180 | 61.1 | 1230 |
| Gemini 1.5-Pro | 71.2 | 180 | **67.4** | 1230 |
| GPT-4o | **72.2** | 180 | 65.3 | **384** |
| **Open-Source MLLM** | | | | |
| LLaVA-OneVision-72B | 62.0 | **32** | 60.0 | 32 |
| Qwen2.5-VL-7B | 65.0 | 180 | - | - |
| Qwen2.5-VL-72B | **76.2** | 180 | - | - |
| VideoChat-A1 (InternVL2.5-8B) | 72.1 | 35.9 | 63.7 | 35.9 |
| VideoChat-A1 (Qwen2.5-VL-7B) | 70.7 | 42 | 64.3 | 42 |
| **Training-Free LLM Agent** | | | | |
| Ours (InternVL2.5-8B) | **74.0** | 11.5 | 66.3 | **20.4** |
| Ours (Qwen2.5-VL-7B) | 72.4 | **10.7** | **67.0** | 23.8 |

cantly fewer frames. This is because our method efficiently captures key frames without relying on global clustering and can better leverage frame-specific information rather than solely depending on captions.

To ensure a fair comparison, we conduct experiments using the same version of the open-source LLM, Mistral-8x7B (Jiang, 2024), as shown in Table 6. The results demonstrate that our method outperforms the state-of-the-art method, DrVideo, in both accuracy and efficiency. This superiority stems from the proposed framework itself, rather than the inherent capability of the underlying LLM.

### 4.3.2 COMPARISON WITH PROPRIETARY AND OPEN-SOURCE MLLMS

We compare our method with state-of-the-art proprietary and open-source MLLMs on EgoSchema and the long split of Video-MME, as shown in Table 2. By incorporating the more powerful VLMs

Table 3: Ablation results of Perception Tools on the NExT-QA subset. C, T, and D denote accuracy (%) on the causal, temporal, and descriptive subsets, respectively.

| Type | DS | TF | SF | C | T | D | Avg. |
|------|------|------|------|------|------|------|------|
| 1 | ✓ | × | × | 74.0 | 63.0 | 79.0 | 72.0 |
| 2 | ✓ | ✓ | × | 77.5 | 66.0 | 79.5 | 74.3 |
| 3 | × | ✓ | ✓ | 73.0 | 62.5 | 80.0 | 71.8 |
| 4 | ✓ | ✓ | ✓ | **79.0** | **67.0** | **84.5** | **76.8** |
| 5 | Uniform-k | ✓ | ✓ | 74.5 | 63.5 | 81.5 | 73.2 |
| 6 | top-k | ✓ | ✓ | 76.5 | 65.5 | 84.0 | 75.3 |

InternVL2.5-8B (Chen et al., 2024) and Qwen2.5-VL-7B (Bai et al., 2025) as components, our method achieves accuracy comparable to Gemini 1.5-Pro and GPT-4o while using only a minimal number of frames, surpassing the standalone Qwen2.5-VL-7B and outperforming VideoChat-A1 (Wang et al., 2025a), which also adopts the same VLMs—InternVL2.5-8B and Qwen2.5-VL-7B—as its components. This performance gain stems from the collaborative design of LLMs and MLLMs, which circumvents the inherent context-window limitations of a single MLLM and enables adaptive, multi-granularity information extraction.

## 4.4 ABLATION STUDY

**Ablation of the Perception Tools.** We perform ablation studies of the proposed Perception Tools on the NExT-QA subset, as shown in Table 3. The results indicate that `divergent search` (DS), `temporal focus` (TF), and `spatial focus` (SF) are complementary. Specifically, DS provides general reasoning cues and consistently benefits all question types. TF is particularly effective for temporal reasoning, such as identifying previous or subsequent events. SF facilitates the recognition of objects, attributes, and counts, which is especially valuable for descriptive questions.

We also compare strategies for `divergent search`, including uniform $k$-frame sampling, similarity-based top-$k$, and our watershed strategy (Types 4–6). Results show that the watershed strategy significantly improves causal and temporal tasks by capturing a broader range of relevant frames. We also provide a visualization analysis of the `divergent search` algorithm, and the details are presented in the Appendix A.2.

Table 4: Ablation results on the agentic framework on EgoSchema.

| Framework | Acc. | Frames |
|------|------|------|
| ReAct (Yao et al., 2023) | 63.2 | 14.7 |
| Ours w/o Verification | 64.8 | **8.1** |
| Ours | **69.2** | 11.2 |

Table 5: Experimental results on Haystack-LVBench. All retrieval metrics are reported in (%).

| Method | Frames | Precision | Recall | F1 |
|------|------|------|------|------|
| VideoAgent* | 10.1 | 1.2 | 8.5 | 2.1 |
| T* (Ye et al., 2025) | 8 | 1.6 | 7.1 | 2.5 |
| Ours w/o Verification | **7.8** | 2.3 | 8.9 | 3.7 |
| Ours | 10.7 | **2.8** | **10.6** | **4.4** |

**Ablation of Agentic Framework.** We evaluate the contribution of the proposed *Perception-Reflection* loop by fixing the toolkit and ablating the framework, as shown in Table 4. Compared with the baseline ReAct (Yao et al., 2023), the Verification-Enhanced Reflection Agent (VERA) with *Evaluation and Planning* substantially reduces the average number of frames (from 14.7 to 8.1), highlighting the key role of Reflection in improving clue discovery efficiency. Moreover, adding *Key Observation Verification* yields a further accuracy improvement of 4.4%, showing that VERA's verification mechanism effectively mitigates the misleading effects of hallucinated information, albeit with an overhead of 3.1 additional frames.

Furthermore, to verify that CogniGPT indeed selects key information and to assess the contribution of the verification mechanism to this process, we conduct a Needle-in-a-Haystack task on Haystack-LVBench (Ye et al., 2025), as shown in Table 5. We evaluate the retrieval performance of task-relevant key frames using Precision, Recall, and F1. The results show that, despite using fewer frames, our retrieval performance is superior, indicating that our method grounds key information more effectively than VideoAgent* and T* (Ye et al., 2025). This advantage arises from our multi-

Table 7: Average runtime, number of LLM calls, and token analysis per sample on EgoSchema. Retrieval includes similarity computation and clustering. Time is in seconds.

| Method | Embedding Time | Retrieval Time | Caption Time | QA Time | LLM | | | Total Time | Frames | Acc. |
| --- | --- | --- | --- | --- | --- | --- | --- | --- | --- | --- |
| | | | | | Time | Calls | Tokens | | | |
| VideoAgent* | 4.0 | 0.2 | 10.8 | - | 84.9 | 10.0 | 1131.0 | 99.9 | **8.4** | 60.2 |
| VideoTree | 4.0 | 0.2 | 80.5 | - | 16.5 | 5.9 | 291.7 | 101.2 | 62.4 | 66.2 |
| DrVideo | 0.6 | 0.1 | 118.5 | 2.4 | 21.7 | 5.0 | 292.0 | 143.3 | >90 | 66.4 |
| Ours | 4.0 | 0.1 | 10.3 | 1.5 | 23.6 | 6.8 | 392.9 | **39.5** | 11.2 | **69.2** |

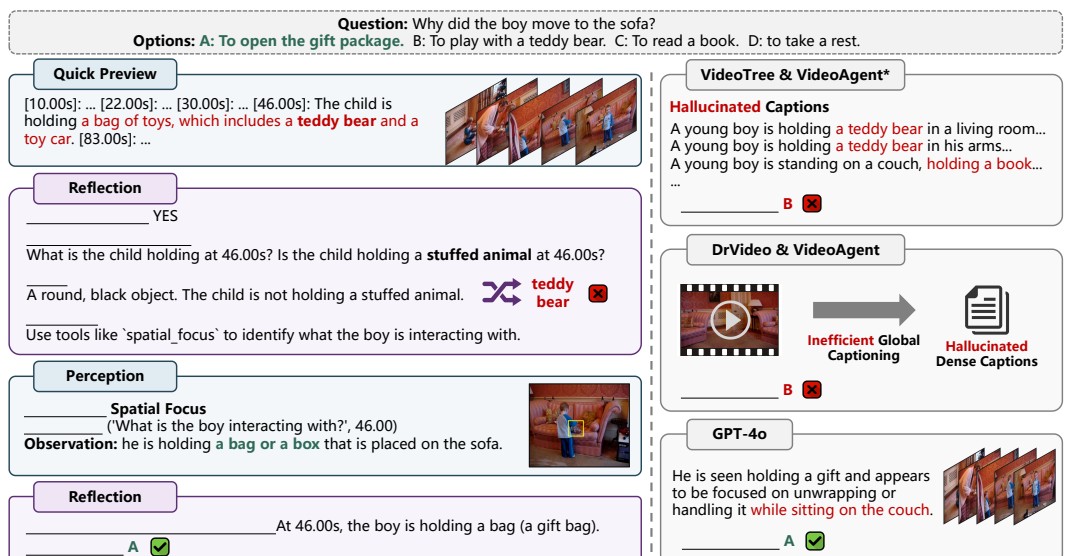

Figure 3: A case study from NExT-QA. CogniGPT progressively explores clues while effectively avoiding interference from hallucinated captions. In contrast, existing training-free LLM agents are misled by such hallucinations, and the global captioning methods of DrVideo and VideoAgent are inefficient. Although GPT-4o provides the correct answer, its reasoning process is affected by hallucinations.

granularity information selection strategy. In addition, the verification mechanism further strengthens key-information grounding by effectively preventing misleading caption hallucinations.

**Ablation of LLMs.** We evaluate our framework with both open-source and proprietary LLMs (Table 6). Results show that our prompting strategy transfers well across different LLMs. While absolute performance varies with the reasoning ability of each LLM, the key contribution lies in the design of the agentic framework.

Table 6: Comparison using an open-source LLM and ablation across open-source and proprietary LLMs on EgoSchema.

| Method | LLM | Acc. | Frames |
| --- | --- | --- | --- |
| DrVideo | Mistral-8x7B | 47.6 | >90 |
| Ours | Mistral-8x7B | **51.8** | 14.9 |
| Ours | GPT-3.5 | 66.2 | 13.3 |
| Ours | DeepSeek-V3 | 68.6 | 12.7 |
| Ours | GPT-4 | **69.2** | 11.2 |

## 4.5 RUNTIME ANALYSIS

To assess the efficiency of our approach, we evaluate the per-sample average runtime, LLM calls, and generated tokens on the EgoSchema dataset using a single NVIDIA A6000 GPU (48GB). The runtime includes video/text embedding, similarity computation/clustering (retrieval), captioning, QA, and LLM responses. Compared to VideoTree and DrVideo, our method is more efficient with fewer caption frames. When compared to VideoAgent*, we further reduce LLM inference time. This efficiency results from our ability to accurately identify key frames and capture their information more comprehensively.

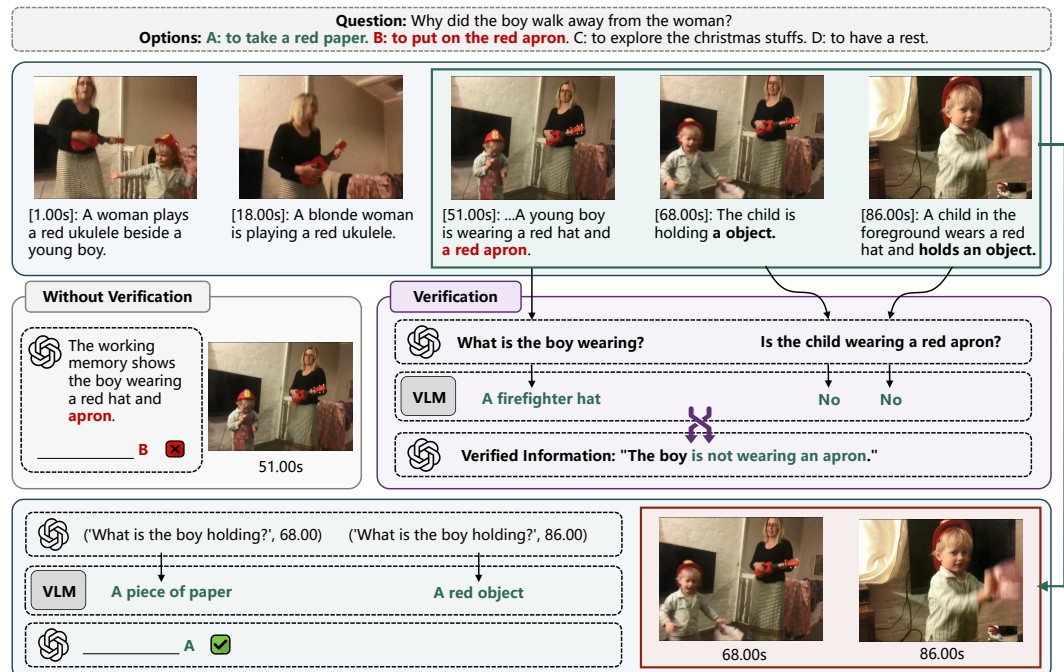

Figure 4: A case demonstrating the process of key information selection and verification. CogniGPT, through the collaboration between the LLM and VLM, eliminates the hallucination "a red apron" in the original caption by performing cross-validation across multiple frames and differently phrased questions. This mechanism further enables CogniGPT to localize the key frame and obtain the correct answer A. In contrast, without the verification mechanism, it incorrectly selects answer B.

## 4.6 CASE STUDY

Figure 3 illustrates a case from the NExT-QA dataset. In this example, CogniGPT first employs the Quick Preview to extract frames depicting the main scene. The Verification-Enhanced Reflection Agent then identifies the frame at 46.00s as a keyframe and detects that the object "teddy bear" is a hallucination. Subsequently, the Multi-Granular Perception Agent performs fine-grained spatial reasoning on the identified keyframe, determining that the true target object is a "bag," and thus selects the correct answer (A). In contrast, existing training-free LLM agents are misled by hallucinated captions, leading them to select (B). Additionally, DrVideo and VideoAgent's dense captioning methods are inefficient. Although GPT-4o provides the correct answer, its reasoning process is affected by hallucinations.

Figure 4 presents a case illustrating the process of key information selection and verification. CogniGPT identifies and mitigates the task-related hallucination "a red apron" in the caption by cross-validating differently phrased questions across multiple candidate key frames, thereby preventing the hallucinated caption from misleading LLM reasoning. This mechanism further enables CogniGPT to accurately localize the true key information and produce reliable answers. We also provide failure cases in Appendix A.3.

## 5 CONCLUSION

In this paper, we propose CogniGPT, a framework that leverages collaborative LLM agents to construct a *Perception-Reflection* loop for temporal visual reasoning. Through this interactive process, CogniGPT progressively explores multi-granular task-relevant cues, enabling efficient and reliable video understanding. Extensive experiments across multiple benchmarks demonstrate that CogniGPT significantly outperforms existing training-free methods in both accuracy and efficiency. We believe this work offers a novel perspective on video understanding and generalizes well to a wide range of dynamic visual reasoning scenarios.

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

# A APPENDIX

## A.1 THE USE OF LARGE LANGUAGE MODELS

We use Large Language Models (LLMs) solely for assisted writing, including spelling and grammar error checking, as well as writing polishing.

## A.2 VISUALIZATION ANALYSIS OF DIVERGENT SEARCH

We conduct a visualization analysis of the `divergent search` algorithm. As shown in Figure 5, unlike the commonly used top-$k$ strategy in VideoAgent (Fan et al., 2024), which often selects highly similar and redundant frames, our watershed strategy produces more diverse and informative frame selections.

Figure 5: Comparison of divergent search strategies on NExT-QA.

## A.3 FAILURE CASE ANALYSIS

We present failure cases in both the perception stage and the verification stage from the EgoSchema dataset, as shown in Figure 6 and Figure 7.

First, we present a failure case in the perception stage, as shown in Figure 6. Specifically, the key to answering the question lies in the pen-marking process occurring between 6.00s and 25.00s, rather than in the cutting process, which ultimately leads to CogniGPT's incorrect response. This failure primarily arises because CogniGPT over-relies on question-relevant segments (e.g., the cutting process) while overlooking potentially causal segments outside the immediately relevant context. When the question and its corresponding answer appear in different portions of the video, such errors are more likely to occur. This observation underscores the need for future research to identify and leverage causally relevant video segments that extend beyond those directly associated with the question.

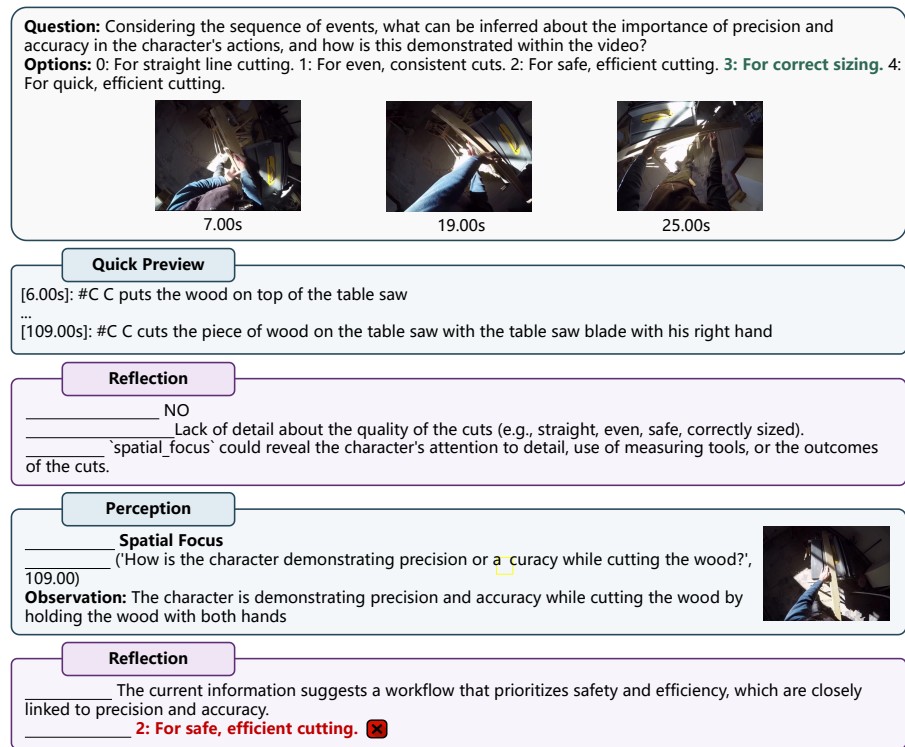

Figure 6: A failure case in the perception stage. The error arises because CogniGPT focuses solely on segments that are directly related to the question (e.g., the cutting process), while overlooking segments that are only potentially related to the question (e.g., the action of marking with a pen).

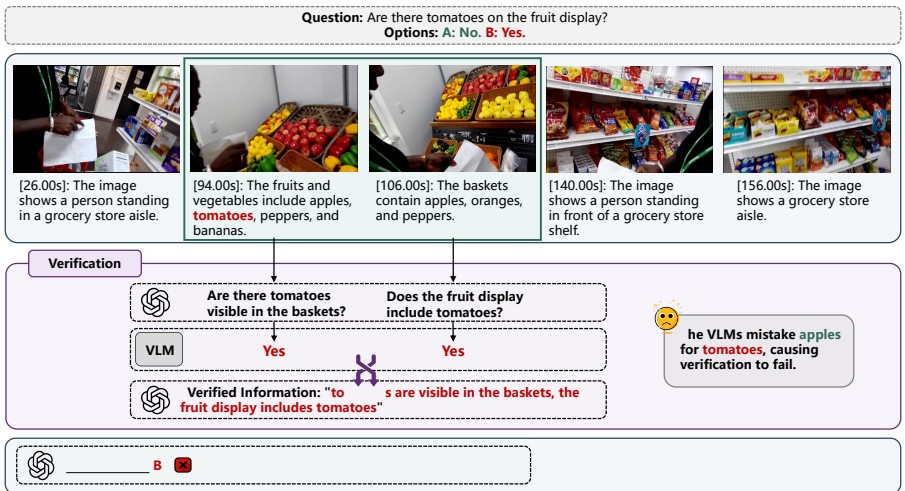

Figure 7: A failure case in the verification stage. The error arises because the VLM is limited by its recognition capability. In extreme cases, the VLM may hallucinate a consistent answer across multiple frames and the question (e.g., repeatedly judging that there are tomatoes in several frames), which leads to verification failure.

Secondly, we present a failure case of the verification stage, as shown in Figure 7. Specifically, for the hallucinated term "tomatoes" in the caption at 94.00s, although multiple frames and questions are used for cross-verification, the VLM's limited recognition capability causes consistent errors across several frames and queries under certain extreme conditions (in this example, the model repeatedly misidentifies an apple as tomatoes across multiple frames). Consequently, the hallucination is not

successfully corrected. Addressing this issue requires improving the perceptual capacity of MLLMs themselves to avoid such consistency-induced hallucinations, which warrants further investigation in future work.

## A.4 PROMPT OF LLM

The LLM prompts for our Reflection Agent, Perception Agent, and perception tools are as follows.

---

**Perception Agent Prompt**

```
You are a Perception Agent responsible for selecting the most
appropriate perception tool to gather information based on the current
understanding state and guidance.
Current question: {question}
Video duration: {video_duration} seconds
Current working memory:
{memory}
Guidance for next action:
{guidance}
Available perception tools:
{tools}
```
**IMPORTANT Notes:**
```
    1. The segment captions with prefix '#C' refer to the camera
       wearer, while those with prefix '#O' refer to someone other
       than the camera wearer.
    2. Do NOT use single letters 'C' or 'O' as query for
       key_frame_selection tool.  Instead, use specific objects,
       actions, or descriptive terms.

You MUST output ONLY the following format with NO additional text or
explanations:
Tool Name:  [selected tool name]
Tool Input:  [tool input parameters]
```

---

**Reflection Agent Step 1: Verification Prompt**

```
You are a Reflection Agent analyzing the latest observation for
task-relevant information.
Question:  {question}
Video duration:  {video_duration} seconds
Latest observation:  {latest_observation}
Note:  '#C' refers to camera wearer, '#O' refers to others.
Analyze the observation:
    1. Does it contain information that is critical for answering the
       question?
    2. If yes, what key information requires verification?  Generate
       two to three distinct verification questions with timestamps,
       to assess whether the key information is hallucinated.
```
**Important Guidelines for Verification Questions:**
```
    • Keep questions SIMPLE and DIRECT – avoid complex or compound
      questions
    • Focus on basic visual facts that can be easily verified (colors,
      objects, actions, positions)

Verification Questions Examples:
Verification Questions:  [("Is the boy's shirt red?", 15.2), ("What
color is the boy's shirt?", 15.2)]
```

---

```
CRITICAL: You MUST output ONLY the following format with NO additional
text or explanations:
Key Information: [YES/NO]

If YES:
Verification Questions: [("question1", timestamp1), ("question2",
timestamp2)]
```

**Reflection Agent Step 2: Sufficiency Assessment Prompt**

```
You are a Reflection Agent. Analyze working memory to determine
information sufficiency.

Current question: {question}
Video duration: {video_duration} seconds

Current working memory:
{working_memory}

Notes:
    1. '#C' = camera wearer, '#O' = other person
    2. All information is verified/reliable
```

**Information Sufficiency Assessment**

```
    1. Information sufficiency: Is collected information sufficient
       to answer reliably?
    2. Information gaps: What key information is missing?
    3. Next step decision: Continue gathering or terminate with
       answer?

Criteria:
    • CONTINUE: Critical info missing, insufficient, low confidence
    • TERMINATE: Sufficient relevant info gathered

Output (be concise):
Analysis: [brief assessment of sufficiency and gaps]
Decision: [continue/terminate]

If CONTINUE:
Guidance: [what to gather next, which tools]

If TERMINATE:
Final Answer: [number 0-4]
```

**Multi-Granular Perception Toolkit**

**1. divergent_search**
Find video segments related to a query within a broad time range and generate rough descriptions.
**TOOL INPUT FORMAT:** ('query_text', (start_time, end_time))
**Input must be:** ('man with glasses', (150.0, 315.0))
**EXAMPLE:** ('person', (0.0, 90.0))
Returns top-k most relevant segments with timestamps and rough descriptions.

**2. spatial_focus**
Analyze spatial relationships and visual attributes at specific time points in the video.
**TOOL INPUT FORMAT:** [('question_text1', time_point1), ('question_text2',
time_point2), ...]
**EXAMPLE:** [('What objects are visible in the scene?', 10.5), ('What
color is the car?', 20.3)]
**NOTE:** Specializes in understanding scene composition, object attributes, and spatial relationships.

**3. temporal_focus**
Identify key scenes within specific time intervals and generate captions.

> **TOOL INPUT FORMAT:** `[(start_time1, end_time1), (start_time2, end_time2), ...]`
> **EXAMPLE:** `[(10.0, 30.0), (37.0, 47.5), (70.0, 78.0)]`
> Returns timestamps and captions of the most representative scenes in the given time ranges.

## A.5 ERROR ANALYSIS

We conduct an error bar analysis on the subsets of the NExT-QA benchmark. Specifically, we run our model 10 times under identical experimental conditions and calculate the mean and standard deviation of accuracy. As shown on the left side of Figure 8, the standard deviations for the causal, temporal, and descriptive questions are 0.2%, 0.3%, and 0.1%, respectively. These results demonstrate that our method exhibits strong robustness and stability across different question types.

Additionally, we perform five experiments with different random seeds on the EgoSchema, Video-MME, NExT-QA, and MovieChat benchmarks. The error bars are shown on the right side of Figure 8. The standard deviations on the four datasets are 0.5%, 0.6%, 0.3%, and 0.3%, respectively. These findings highlight the robustness and generalization ability of our method across datasets with diverse contexts and temporal scales, ensuring the statistical significance and reliability of the reported results.

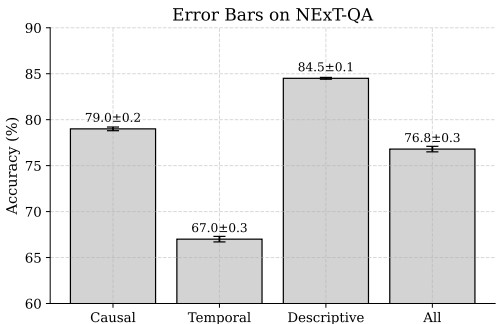 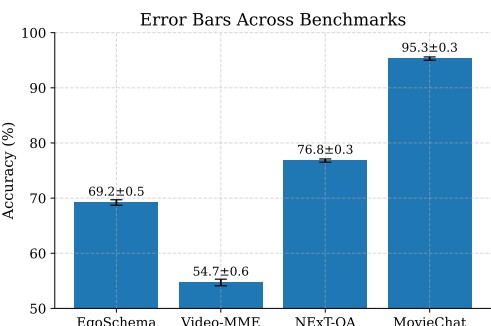

Figure 8: Error bar analysis across benchmarks. Left: Error bar analysis on the NExT-QA benchmark. C, T, D, and All denote the accuracy (%) on the causal, temporal, and descriptive subsets, and their overall average, respectively. Right: Error bars across EgoSchema, Video-MME, NExT-QA, and MovieChat benchmarks.

## A.6 DATASET DETAILS

We conduct experiments on four video understanding benchmarks: EgoSchema (Mangalam et al., 2023), Video-MME (Fu et al., 2024), NExT-QA (Xiao et al., 2021), and MovieChat (Song et al., 2024).

**EgoSchema** contains first-person perspective videos (avg. 3 min) of daily activities, with multiple-choice questions requiring temporal understanding. We use the official subset of 500 labeled samples for fair comparison.

**Video-MME** is a recently introduced benchmark for evaluating MLLMs on comprehensive video understanding. In our experiments, we focus on its long-form subset, which consists of 300 open-domain videos (30–60 min, avg. 41 min) annotated with 900 expert-crafted multiple-choice questions. This subset emphasizes long-range temporal reasoning over extended video contexts.

**NExT-QA** assesses temporal reasoning, causal analysis, and descriptive understanding, with videos averaging 44 seconds. We sample 600 instances (200 from each of the three question categories) for evaluation.

**MovieChat** consists of 1,000 movie/TV clips (average length: 10 minutes). We use the official test split of 170 samples and evaluate using the *Global Mode*, which performs holistic analysis.

The datasets used in this experiment cover a wide range of video durations, question types, and reasoning tasks, ensuring a comprehensive evaluation of CogniGPT.

