# OpenReview forum: "Perceive, Reflect and Understand Long Video: Progressive Multi-Granular Clue Exploration with Interactive Agents"
_ICLR.cc/2026/Conference — Submitted to ICLR 2026_

### Official Review · Reviewer_ha9r · 2025-10-24

**Soundness:** 2
**Presentation:** 2
**Contribution:** 2
**Rating:** 4
**Confidence:** 4

**Summary:**

The paper introduces a training-free agent for long-form video QA that alternates between perception and verification in a loop manner. For perception, it gathers the information via 1) a diversity-promoting keyframe selector; 2) temporal clustering within segments; 3) spatial VQA on selected frames. For verification, it evaluates key hypotheses using targeted VQA to decide whether to continue the search or provide an answer. On EgoSchema, Video-MME, NExT-QA, and MovieChat, the method achieves improved accuracy with fewer processed frames than agentic baselines.

**Strengths:**

- Sound framework. A simple yet working training-free agent framework that alternates between perception and verification, reflecting a natural way for long-form video understanding.

- Good results. It shows that the system, CogniGPT, achieves good results on multiple video QA benchmarks, including EgoSchema, Video-MME, NExT-QA, and MovieChat.

- Useful ablations. The paper ablates how components like LLM and the verification stage affect the final performance.

**Weaknesses:**

- Limited novelty. The training-free agentic framework in video understanding has been well explored, for example, in VideoAgent. Recent papers have already studied training-based agents that achieve better performance, such as VideoR1. This paper follows the VideoAgent, with perception using a retrieval system (already proposed by VideoAgent) and reflection and verification using an LLM (also suggested by VideoAgent), in a loop (again proposed by VideoAgent). Given the limited novelty, more insightful analysis, such as failure cases when applying the method to this setting, the reasons for the failures, and how to adapt them, is worth further study.

- More ablations needed. One novelty from the paper is the "diversity-promoting" keyframe selector. However, there is no ablation showing how this diversity helps the final performance. Meanwhile, it needs more analysis of where the errors occur, both at the perception stage and at the verification stage.

- Statistical significance. The error bars are only provided for NExT-QA; please add confidence intervals/seed variance for main tables or at least EgoSchema and Video-MME.

**Questions:**

See weaknesses.

Typos: many \cite should be replaced as \citep.

---

> ### Author Response · Authors · 2025-11-23
> **Response to Reviewer ha9r (Part 1 of 2)**
>
> We sincerely appreciate your valuable comments and constructive feedback. Following your suggestions, we address all raised questions, add supplementary experiments and visualizations, and revise the manuscript to improve clarity and rigor. We truly hope that these clarifications and improvements help resolve your concerns, and we respectfully ask that you kindly reconsider our work.
>
> **Q1:** Limited novelty. The training-free agentic framework in video understanding has been well explored, for example, in VideoAgent. Recent papers have already studied training-based agents that achieve better performance, such as VideoR1. This paper follows the VideoAgent, with perception using a retrieval system (already proposed by VideoAgent) and reflection and verification using an LLM (also suggested by VideoAgent), in a loop (again proposed by VideoAgent).
>
> **A1:** Thank you for your insightful comment. While prior work such as VideoAgent has explored agentic frameworks for video understanding, **our method departs from these approaches in several fundamental and essential aspects.**
>
> **1. Perception of task-relevant information.**
>
> First, we differ fundamentally in **how task-relevant information is perceived and selected**. Our framework integrates MLLMs to extract visual information at **arbitrary temporal and spatial granularity**, whereas existing approaches rely solely on a retrieval-caption pipeline that provides only single-granularity localization**. As a result, **prior methods are highly dependent on retrieval accuracy and caption quality**. In contrast, our approach **extracts visual information at arbitrary temporal and spatial granularity**, thereby **avoiding** the missed evidence introduced by fixed-granularity retrieval and the limited detail provided by caption-only descriptions. As demonstrated by the ablation results in Table 3 (Type 1–4), our multi-granularity perception strategy substantially outperforms its single-granularity counterparts.
>
> **Table 3. Ablation results of Perception Tools on the NExT-QA subset. C, T, and D denote accuracy (%) on the causal, temporal, and descriptive subsets, respectively.**
>
> | Type | DS | TF | SF | C    | T    | D    | Avg. |
> |------|----|----|----|-------|-------|-------|-------|
> | 1    | ✓  | ×  | ×  | 74.0 | 63.0 | 79.0 | 72.0 |
> | 2    | ✓  | ✓  | ×  | 77.5 | 66.0 | 79.5 | 74.3 |
> | 3    | ×  | ✓  | ✓  | 73.0 | 62.5 | 80.0 | 71.8 |
> | 4    | ✓  | ✓  | ✓  | **79.0** | **67.0** | **84.5** | **76.8** |
>
> **2.Reflection and verification.**
>
> Second, we also differ fundamentally in **reflection and verification**. Existing approaches rely solely on captions for reasoning, and **once the captions are hallucinated, they cannot be corrected**. In contrast, our method employs a Verification-Enhanced Reflection Agent that collaborates with MLLMs: it queries **multiple frames with different questions** to obtain **complementary information**, essentially performing **cross-view and cross-granularity verification to correct caption hallucinations**. In this way, we greatly reduce the misleading influence of caption hallucinations on the reasoning process.
>
> To illustrate this more clearly, we add a case study in Figure 4 that shows how the verification mechanism corrects the reasoning process. In addition, we include a Needle-in-a-Haystack experiment on Haystack-LVBench [1] in Table 5, which demonstrates that our verification module improves key-frame retrieval performance. Table 4 further shows that introducing our correction mechanism effectively improves the final reasoning accuracy.
>
> **Table 5. Experimental results on Haystack-LVBench. All retrieval metrics are reported in (%).**
> | Method                | Frames | Precision | Recall | F1  |
> |-----------------------|--------|-----------|--------|-----|
> | VideoAgent*           | 10.1   | 1.2       | 8.5    | 2.1 |
> | T*                    | 8      | 1.6       | 7.1    | 2.5 |
> | Ours w/o Verification | **7.8**| 2.3       | 8.9    | 3.7 |
> | Ours                  | 10.7   | **2.8**   | **10.6** | **4.4** |
>
> **Table 4. Ablation results on the agentic framework on EgoSchema.**
>
> | Framework             | Acc.  | Frames |
> |-----------------------|-------|--------|
> | ReAct (Yao et al., 2023) | 63.2  | 14.7   |
> | Ours w/o Verification | 64.8  | **8.1** |
> | Ours                  | **69.2** | 11.2  |
>
> It is worth noting that, under existing paradigms, MLLMs are difficult to incorporate directly. To better address video understanding, we therefore design a new framework that explicitly integrates MLLMs for perception at arbitrary temporal and spatial granularity, as well as for cross-verification of hallucinations.
>
> [1] Re-thinking temporal search for long-form video understanding, CVPR 2025.

---

> ### Author Response · Authors · 2025-11-23
> **Response to Reviewer ha9r (Part 2 of 2)**
>
> **Q2:** Given the limited novelty, more insightful analysis, such as failure cases when applying the method to this setting, the reasons for the failures, and how to adapt them, is worth further study. Meanwhile, it needs more analysis of where the errors occur, both at the perception stage and at the verification stage.
>
> **A2:** Thank you for your valuable suggestion. Following your advice, we add failure cases and analyses for both the **perception stage** and the **verification stage**, as illustrated in Figures 6 and 7 in Appendix A.3.
>
> At the **perception stage**, errors tend to occur when **the question and its corresponding answer appear in different video segments**. Our framework prioritizes segments that are directly relevant to the question, which may lead to partial neglect of latent causally related segments. The failure arises from **the model’s insufficient ability to infer causal connections between the question and the underlying visual evidence**. Addressing this issue requires future work on explicitly modeling the causal structure between queries and observations, and on identifying video regions that are causally related to the answer even when not directly referenced by the question.
>
> At the **verification stage**, our method may fail in **rare cases** where multiple frames from different viewpoints and multiple questions all exhibit **consistent hallucinations** (e.g., when MLLMs repeatedly confuse “apples” with “tomatoes”). The failure occurs because, in such extreme situations, **the MLLM’s limited recognition capability** leads it to produce similar erroneous outputs across different frames and queries, making it difficult for the verification step to distinguish hallucinated content. Nevertheless, our verification mechanism, which leverages collaboration between the LLM and the MLLM and performs cross-verification across multiple viewpoints, still remains **an effective strategy for mitigating hallucinations**. As shown in Tables 4 and 5 and Figure 4, this strategy effectively corrects single-granularity caption hallucinations and improves both key-frame retrieval and reasoning performance.
>
> In summary, these failure modes highlight important directions for future research, and we greatly appreciate the reviewer’s suggestion to deepen the analysis in these aspects.
>
> **Q3:** More ablations needed. One novelty from the paper is the "diversity-promoting" keyframe selector. However, there is no ablation showing how this diversity helps the final performance.
>
> **A3:** Thank you for the helpful comment. We believe that the “diversity-promoting” keyframe selector you refer to corresponds to our proposed divergent-search mechanism inspired by the watershed concept. **In fact, Table 3 already demonstrates that this diversity improves the final performance by comparing Type-4 to Type-6 settings**. Specifically, we compare our divergent-search strategy (implemented via the watershed algorithm) with Uniform-k and Top-k baselines, and we show that the introduced diversity consistently benefits downstream performance, especially on causal and temporal reasoning tasks.
>
> This improvement arises because our divergent-search strategy is **deliberately designed to exploit the semantic structure of video sequences**. Adjacent frames are often redundant due to high semantic similarity, whereas **temporally distant yet semantically related frames frequently capture the causal dependencies** essential for reasoning. Our method explicitly selects such causally informative frames over a broader temporal span, enabling more complete contextual understanding and stronger inference. By contrast, traditional Top-k selectors tend to focus on nearby redundant frames, leading to limited contextual coverage.
>
> **Table 3: Ablation results on the NExT-QA subset.**
> | Type | DS | C    | T    | D    | Avg. |
> |------|----|-------|-------|-------|-------|
> | 4 | watershed | **79.0** | **67.0** | **84.5** | **76.8** |
> | 5 | Uniform-k | 74.5 | 63.5 | 81.5 | 73.2 |
> | 6 | top-k     | 76.5 | 65.5 | 84.0 | 75.3 |
>
>
> **Q4:** Statistical significance. The error bars are only provided for NExT-QA; please add confidence intervals/seed variance for main tables or at least EgoSchema and Video-MME.
>
> **A4:** Thank you for your valuable suggestion. Following your advice, we conduct five runs with different random seeds on EgoSchema, Video-MME, NExT-QA, and MovieChat, and we include the corresponding error bars and analysis in Figure 8 of Appendix A.5. The standard deviations on these four datasets are 0.5%, 0.6%, 0.3%, and 0.3%, respectively. These results demonstrate the robustness and generalization ability of our method across datasets with diverse contexts and temporal scales, ensuring the statistical significance and reliability of our findings.
>
> **Q5:** Typos: many \cite should be replaced as \citep.
>
> **A5:** Thank you for pointing this out. We have replaced all occurrences of \cite with \citep throughout the manuscript.

---

### Official Review · Reviewer_mkrX · 2025-10-26

**Soundness:** 3
**Presentation:** 3
**Contribution:** 2
**Rating:** 4
**Confidence:** 4

**Summary:**

CogniGPT presents a training-free agentic framework for long-video question answering that addresses the computational inefficiency of dense-captioning approaches through progressive multi-granular exploration. The system comprises two coordinated agents operating iteratively with a maximum of three rounds. The Multi-Granular Perception Agent selectively employs three complementary tools that capture evidence at different granularities: Divergent Search uses EVA-CLIP similarity with watershed-style peak detection to identify salient keyframes across the full video, Temporal Focus applies K-means clustering within relevant spans to capture sub-event structures, and Spatial Focus conducts targeted frame-level visual question answering for fine-grained attributes and spatial relationships that captions may miss. The Verification-Enhanced Reflection Agent mitigates hallucination through explicit cross-validation of key observations by auto-generating verification questions tied to specific timestamps, while also evaluating information sufficiency and planning subsequent exploration steps.

**Strengths:**

* Modular, training-free stack with clear swap points. The MGPA/VERA split isolates retrieval, captioning, and VQA so backbones (EVA-CLIP, captioner, VLM) can be upgraded without controller retraining—useful for rapid iteration and ablations.
* Strong accuracy-per-frame trade-off. Consistently competitive results while processing only ~10–18 frames/sample on long-video suites, indicating the coarse-to-fine tool cascade is compute-efficient relative to dense captioning.

**Weaknesses:**

* Limited conceptual novelty. The method mainly recombines established pieces—agentic plan-act loops (VideoAgent family), hierarchical/multi-granular retrieval (e.g., VideoTree-style coarse→fine), dense-caption→LLM aggregation (LLoVi/DrVideo), and Chain-of-Verification—plus an engineering tweak (watershed peak picking).

**Questions:**

Could the authors provide evidence that the verification loop truly improves grounding accuracy, rather than merely promoting self-consistency among correlated VLMs?

---

> ### Author Response · Authors · 2025-11-23
> **Response to Reviewer mkrX (Part 1 of 3)**
>
> We sincerely appreciate your valuable comments and thoughtful feedback. Following your suggestions, we address all questions, add supplementary experiments and visualizations, and revise the manuscript to improve clarity and rigor. We truly hope that these clarifications and improvements help resolve your concerns, and we respectfully ask that you kindly reconsider our work.
>
> **Q1:** Limited conceptual novelty. The method mainly recombines established pieces—agentic plan-act loops (VideoAgent family), hierarchical/multi-granular retrieval (e.g., VideoTree-style coarse→fine), dense-caption→LLM aggregation (LLoVi/DrVideo), and Chain-of-Verification—plus an engineering tweak (watershed peak picking).
>
> **A1:** Thank you for your insightful comment. We acknowledge that some related concepts appear in prior work, **but our method differs from them in several fundamental ways:**
>
> **1. Difference from plan–act loops (e.g., VideoAgent)**
>
> Prior agentic plan–act loops essentially operate on **single-granularity relevant-frame retrieval followed by captioning**. In these methods, the LLM decides the retrieval parameters, and the system repeatedly performs similarity-based retrieval and caption generation. Thus, the loop is primarily driven by retrieval sufficiency, and it inherits two inherent limitations: **(a) strong dependence on retrieval accuracy**, and **(b) strong dependence on caption quality**, which may still **lead to confident yet incorrect answers when hallucinated captions appear adequate**.
>
> In contrast, our loop fundamentally focuses on **multi-granularity extraction of reliable cues**. The LLM dynamically determines how and at what granularity to perceive the video, and collaborates with MLLMs to validate information from multiple viewpoints. Our loop is therefore driven by **information reliability, rather than retrieval sufficiency**. This shift substantially reduces dependence on both retrieval performance and caption fidelity. It enables the system to uncover and correct information across granularities, allowing video understanding with the minimum amount of reliable evidence.
>
> Our experiments confirm this point, and the results in Table 1 and Table 4 show that our loop achieves better accuracy with fewer frames than traditional agentic loops such as VideoAgent and ReAct.
>
> **Table 1&4: Results on EgoSchema.**
> | Framework                 | Acc. | Frames |
> |---------------------------|------|--------|
> | VideoAgent        | 60.2 | 8.4   |
> | ReAct            | 63.2 | 14.7   |
> | VideoTree        | 66.2 | 62.4   |
> | LLoVi        | 51.8 |  180   |
> | DrVideo        | 66.4 |  >90   |
> | Ours                  | **69.2** |  11.2  |
>
> **2.Difference from hierarchical / multi-granular retrieval (VideoTree-style)**
>
> Existing hierarchical retrieval methods are **essentially clustering-driven**: the hierarchy is constructed based on frame similarity, and their performance strongly **depends on the effectiveness of the clustering algorithm**.
> Our hierarchical/multi-granular mechanism is instead **LLM-driven**. The LLM continuously **designs perception strategies and granularity levels** according to the evolving context. This essential difference allows our system to more efficiently and accurately capture temporally and semantically relevant multi-granular information, rather than relying on frame-similarity structures.
>
> Our experiments confirm this point, and the results in Table 1 show that our approach achieves higher accuracy than VideoTree while using significantly fewer frames, benefiting from the LLM-driven hierarchical strategy rather than a clustering-driven one.
>
> **3.Difference from dense-caption → LLM aggregation (LLoVi/DrVideo)**
>
> Dense-captioning methods **convert the entire video into single-granularity captions**. In contrast, our method uses the LLM to plan the perception strategy and scope—**avoiding redundant processing of the entire video**—and employs multiple models to **extract multi-granular information**.
>
> Benefiting from this fundamental difference, our method is **less affected by redundant single-granularity noisy captions** and **requires processing only a small number of frames**, resulting in higher accuracy and efficiency.
>
> Our experiments confirm this point, and Table 1 shows that our method achieves substantially higher accuracy than LLoVi and DrVideo while requiring far fewer frames.

---

> ### Author Response · Authors · 2025-11-23
> **Response to Reviewer mkrX (Part 2 of 3)**
>
> **4. Difference from Chain-of-Verification (CoVe)**
>
> The underlying principle of CoVe [1] is that LLMs exhibit lower hallucination rates when generating short texts compared to long ones. In contrast, our verification mechanism is **fundamentally different**. It relies on **the complementary multi-view and multi-granularity visual information captured by the MLLM from different frames, enabling cross-verification to remove hallucinations**.
>
> To clearly illustrate this distinction, we add Figure 4 to show our verification process. In this example, a hallucinated caption such as “red apron” arises when observing only a single frame, but cross-verification with adjacent frames from different viewpoints confirms that the object is actually a red piece of paper rather than an apron. This correction prevents the hallucinated caption from misleading the reasoning process. Such a mechanism is fundamentally different from CoVe.
>
> **5. Difference from watershed-style peak selection**
>
> Our method is **fundamentally different** from the classical watershed algorithm used in digital image processing. Although **we borrow only the conceptual metaphors of “peaks” and “valleys,”** the underlying mechanism is entirely unrelated. Traditional watershed is a gradient-based region-growing technique that segments spatial regions by following topographic contours, whereas our method performs no region growing at all. Instead, it essentially serves as a **semantic-density peak selection process** that identifies task-relevant key frames along the temporal dimension. Thus, it is not connected to the principles, goals, or operational behavior of the conventional watershed algorithm.
>
> Importantly, this design is not a simple engineering tweak but **a mechanism deliberately constructed around the semantic characteristics of video data**. Adjacent frames are often highly redundant and share nearly identical semantics, whereas non-adjacent but semantically related frames frequently encode causal or task-relevant events. Our strategy explicitly captures these patterns. As demonstrated by the Type 4–6 ablation results in Table 3, the proposed mechanism effectively identifies long-range dependencies and causally meaningful frames, highlighting its contribution to the overall performance.
>
> **Table 3: Ablation results on the NExT-QA subset.**
> | Type | Divergent Search | C    | T    | D    | Avg. |
> |------|---------|-------|-------|-------|-------|
> | 4 | watershed | **79.0** | **67.0** | **84.5** | **76.8** |
> | 5 | Uniform-k | 74.5 | 63.5 | 81.5 | 73.2 |
> | 6 | top-k     | 76.5 | 65.5 | 84.0 | 75.3 |
>
> [1] Chain-of-verification reduces hallucination in large language models, ACL 2024.

---

> ### Author Response · Authors · 2025-11-23
> **Response to Reviewer mkrX (Part 3 of 3)**
>
> **Q2:** Could the authors provide evidence that the verification loop truly improves grounding accuracy, rather than merely promoting self-consistency among correlated VLMs?
>
> **A2:** Thank you for raising this important question. We provide both **quantitative** and **qualitative** evidence demonstrating that the verification loop truly improves grounding accuracy, rather than simply encouraging self-consistency among correlated VLMs.
>
> **1. Quantitative evidence.**
>
> We conduct a **Needle-in-a-Haystack task** on Haystack-LVBench [2], which evaluates key-frame grounding performance in long videos using Precision, Recall, and F1. The results, presented in Table 5, show a clear degradation in retrieval performance when the verification mechanism is removed. This indicates that our verification loop indeed improves grounding accuracy. The improvement stems from its ability to correct hallucinated caption information and prevent such noise from misleading the key-frame selection process.
>
> **Table 5. Experimental results on Haystack-LVBench. All retrieval metrics are reported in (%).**
> | Method                | Frames | Precision | Recall | F1  |
> |-----------------------|--------|-----------|--------|-----|
> | VideoAgent*           | 10.1   | 1.2       | 8.5    | 2.1 |
> | T*                    | 8      | 1.6       | 7.1    | 2.5 |
> | Ours w/o Verification | **7.8**| 2.3       | 8.9    | 3.7 |
> | Ours                  | 10.7   | **2.8**   | **10.6** | **4.4** |
>
> **2. Qualitative evidence.**
>
> We also provide a case study illustrating how the verification mechanism assists in key-frame selection, as shown in Figure 4. Through cooperation between the LLM and the VLM, cross-validation is performed across multiple related frames by leveraging their complementary multi-view information. This process **filters out frames containing misleading content** (e.g., the frame at 51s) and further **narrows down the candidate key-frame range** (e.g., the frames at 68s and 86s), allowing subsequent reasoning to operate on more reliable visual evidence.
>
> Fundamentally, this mechanism exploits the complementary multi-granularity cues captured by the VLM from different viewpoints, overcoming the limitations of single-view, single-granularity perception. It is not merely a form of self-consistency among VLMs, but rather **a principled way of leveraging multi-view complementarity, which truly improves grounding accuracy**.
>
> [2] Re-thinking temporal search for long-form video understanding, CVPR 2025.

---

### Official Review · Reviewer_j8ZP · 2025-10-26

**Soundness:** 4
**Presentation:** 4
**Contribution:** 3
**Rating:** 8
**Confidence:** 4

**Summary:**

The authors propose CogniGPT, a framework that employs an interactive Perception-Reflection loop between two LLM agents: the Multi-Granular Perception Agent (MGPA) and the Verification-Enhanced Reflection Agent (VERA). MGPA uses tools like divergent and focused search to selectively extract minimal, multi-granular clues from the video, bypassing the need for computationally expensive global video processing. VERA then verifies the reliability of these clues to mitigate hallucinations and guides MGPA's next perceptual step, ensuring an iterative and progressive understanding process. The authors perform experiments across several benchmarks and demonstrate CogniGPT’s superior accuracy and efficiency compared to existing training-free LLM agent methods and competitive performance against proprietary multimodal LLMs.

**Strengths:**

- The main strength of this paper's approach is its ability to achieve high accuracy and significant efficiency for understanding long videos.
- Another strength is the method's improved reliability in reasoning, especially concerning hallucinations introduced by VLMs, thanks to VERA.
- The introduced MGPA captures information at different levels of detail, ensuring that both the broad context and fine details are acquired
- The proposed framework achieves SOTA results across various long video understanding tasks and datasets (EgoSchema, Video-MME, NExT-QA, and MovieChat)

**Weaknesses:**

- A minor complaint I have is that CogniGPT falls into the category of LLM Agent-based methods, and its entire architecture is built around LLM agents, which is the paradigm used by Llovi, VideoTree, and others which might hurt its perceived novelty.
- Minor typos: Figure 2, within the Spatial Focus box, “**Vison**-Language Model”

**Questions:**

- What’s the difference between Multi-Granular Perception Toolkit and Multi-Granular Perception Agent? Should we stick to one terminology?

---

> ### Author Response · Authors · 2025-11-23
> **Response to Reviewer j8ZP**
>
> We sincerely appreciate your positive assessment of our work. Following your suggestions, we address all raised questions and revise the manuscript accordingly in order to resolve potential concerns.
>
> **Q1:** A minor complaint I have is that CogniGPT falls into the category of LLM agent-based methods, and its entire architecture is built around LLM agents, which is the paradigm used by Llovi, VideoTree, and others, which might hurt its perceived novelty.
>
> **A1:** Thank you for your insightful comment. Compared with existing LLM agent–based methods, the novelty of our approach lies in leveraging the collaboration between the LLM and the MLLM to avoid the traditional reliance on single-granularity captioning.
>
> First, our method fundamentally differs in **how it obtains key information**. Instead of relying on fixed-granularity captions (as in Llovi’s dense captioning or VideoTree’s clustering-based retrieval), our approach uses the LLM’s planning and reasoning ability to adaptively determine both the perception range and granularity. This leads to higher efficiency and avoids dependence on auxiliary retrieval performance.
>
> Second, our method fundamentally differs in **how it reflects on and verifies key information**. Through LLM–MLLM collaboration, our method performs cross-validation using complementary information from multiple frames and multiple granularities, which allows it to correct hallucinations in captions. In contrast, existing paradigms rely solely on single-granularity captions, causing their reasoning to heavily depend on caption quality and making them highly vulnerable to caption hallucinations.
>
> Overall, our approach exploits the complementary strengths of the LLM (reasoning and planning) and the MLLM (perception), forming a new paradigm that enables multi-granularity key information extraction and verification, ultimately achieving more reliable and comprehensive long video understanding.
>
> **Q2:** Minor typos: Figure 2, within the Spatial Focus box, “Vison-Language Model”.
>
> **A2:** Thank you for your correction. We have fixed the typo in Figure 2 and updated it to “Vision-Language Model.”
>
> **Q3:** What’s the difference between Multi-Granular Perception Toolkit and Multi-Granular Perception Agent? Should we stick to one terminology?
>
> **A3:** Thank you for seeking this clarification. The Multi-Granular Perception Toolkit (MGPT) serves as the action space of the Multi-Granular Perception Agent (MGPA). Specifically, at each iteration, MGPA selects an action—that is, a tool from the MGPT—along with its corresponding input based on the current context and policy.
>
> We have clarified this terminology relationship in Section 3.1 Overview, where we state:
> “Specifically, at each iteration $t$, MGPA $\pi_p$ selects an action $a_t$ (i.e., a tool from the \textit{Multi-Granular Perception Toolkit}) and its corresponding input based on the query $Q$ ...”
>
> To avoid confusion, we ensure consistent terminology throughout the revised manuscript.

---

> ### Comment · Reviewer_j8ZP · 2025-11-24
>
> I want to thank the authors for their answer. After reading all the other reviewers comments and the authors answers, I have decided to keep my positive rating.

---

> ### Author Response · Authors · 2025-12-01
>
> Dear Reviewer j8ZP,
>
> We sincerely appreciate that you took the time to read the other reviewers’ comments and our rebuttal in full. We are grateful that you believed that our responses have addressed the concerns raised in the reviews, and that you maintained your positive rating. We are also glad that our clarifications have addressed your concerns regarding novelty and terminology. Thanks again for your valuable comments, which have helped us improve our paper.

---

### Official Review · Reviewer_m7vg · 2025-10-31

**Soundness:** 2
**Presentation:** 1
**Contribution:** 1
**Rating:** 2
**Confidence:** 4

**Summary:**

This paper tackles the challenge of understanding complex long videos with sparse task-relevant information. To improve AI reasoning, the paper propose CogniGPT, a framework inspired by human visual cognition. CogniGPT combines a perception agent (MPGA) to focus on task-relevant details, and a reflection agent (VERA)  to verify key information, reducing errors and improving efficiency.

**Strengths:**

- The paper is well-written and easy to follow

- The figures are intuitive.

**Weaknesses:**

- The proposed method is straightforward and intuitive.

 - As an LLM-based agent, its performance is inferior to single models like Qwen2.5-VL and significantly worse than VideoChat-A1[1], which uses VL models. This raises the question of why such a complex LLM agent is needed when simpler and more effective alternatives exist.

- While it uses fewer frames, the claim of "selecting key information" is not convincingly supported by the results.

- The table formatting in the paper is poorly organized.

- The paper should use \citep instead of \cite for citation


[1] Wang Z, Chen B, Yue Z, et al. VideoChat-A1: Thinking with Long Videos by Chain-of-Shot Reasoning[J]. arXiv preprint arXiv:2506.06097, 2025.

**Questions:**

See weaknesses above.

---

> ### Author Response · Authors · 2025-11-23
> **Response to Reviewer m7vg (Part 1 of 2)**
>
> We sincerely appreciate your valuable comments and constructive feedback. Following your suggestions, we address all raised questions, add supplementary experiments and visualizations, and revise the manuscript to improve clarity and readability. We genuinely hope that our clarifications and improvements encourage you to kindly reconsider our work.
>
> **Q1:** As an LLM-based agent, its performance is inferior to single models like Qwen2.5-VL and significantly worse than VideoChat-A1[1], which uses VL models. This raises the question of why such a complex LLM agent is needed when simpler and more effective alternatives exist.
>
> **A1:** Thank you for your valuable feedback. We would like to clarify that, actually, **our method outperforms both Qwen2.5-VL-7B and VideoChat-A1 [1]**. As shown in Table 2, besides the original EgoSchema results, we additionally report Video-MME and include a direct comparison with VideoChat-A1. When using the same strong VLMs (InternVL2.5-8B and Qwen2.5-VL-7B), our framework consistently surpasses both standalone VLMs and VideoChat-A1 on EgoSchema and Video-MME, while using far fewer frames. It is important to note that **Table 1 uses weaker VLMs as components in order to ensure a fair comparison** with existing training-free agents, which are built on the same backbones. The improvements mainly stem from our framework’s ability to let the LLM adaptively plan the perception strategy, range, and granularity, and correct VLM hallucinations through cross-verification across frames—capabilities that a single VLM does not offer on its own. Since stronger VLMs such as Qwen2.5-VL-7B perform better on benchmarks with complex visual elements like Video-MME, integrating them into our framework further boosts performance and yields results that significantly surpass the VLM itself. This indicates that our framework effectively enhances the video understanding ability of arbitrary VLMs without any additional training.
>
> **Table 2. Comparison with Proprietary and Open-Source MLLMs on EgoSchema and Video-MME long split.**
> | Method                         | EgoSchema Acc. | EgoSchema Frames | Video-MME Acc. | Video-MME Frames |
> |--------------------------------|----------------|------------------|----------------|------------------|
> | Qwen2.5-VL-7B                  | 65.0           | 180              | -              | -                |
> | VideoChat-A1 (InternVL2.5-8B)  | 72.1           | 35.9             | 63.7           | 35.9             |
> | VideoChat-A1 (Qwen2.5-VL-7B)   | 70.7           | 42               | 64.3           | 42               |
> | Ours (InternVL2.5-8B)          | **74.0**       | 11.5             | 66.3           | **20.4**         |
> | Ours (Qwen2.5-VL-7B)           | 72.4           | **10.7**         | **67.0**       | 23.8             |
>
> From a methodological perspective, the core purpose of our LLM-based agent is to **address the inherent limitations of single VLMs** in long video understanding. Due to constrained context windows, a single VLM typically resorts to uniformly sparse frame sampling, which often overlooks task-relevant information. In contrast, our approach leverages the decision-making capabilities of an LLM agent to perform adaptive, multi-granularity information extraction, which brings several key advantages:
>
> **1.Adaptive multi-granularity perception.** First, our method achieves adaptive multi-granularity perception. By leveraging the planning and reasoning capabilities of the LLM agent, it adaptively determines the perception strategy and granularity based on the difficulty of the query and the current context, and thereby focuses on task-relevant information. This behavior is fundamentally different from Qwen2.5-VL and VideoChat-A1, which rely on a fixed, single-granularity perception scheme.
>
> **2.Verification and mitigation of VLM hallucinations.** Benefiting from the reasoning capability of the LLM agent, our method can perform cross-verification over complementary information from different viewpoints across multiple frames, and correct hallucinations of the VLM, which a single VLM cannot achieve. We provide an additional illustration of this correction process in Figure 4. Moreover, we include supplementary experiments on the Needle-in-a-Haystack task in Table 5, which further demonstrate that our verification mechanism substantially enhances the grounding of key frames.
>
> **3.Higher efficiency.** Benefiting from the LLM agent deciding the strategy, granularity, and range for perceiving key frames, our method avoids processing the entire video. Instead, it focuses on a substantially smaller subset of informative frames, resulting in significantly improved efficiency. As shown in Table 2, our approach achieves better performance while using markedly fewer frames, highlighting the high efficiency of the LLM-driven selection strategy.
>
> [1] VideoChat-A1: Thinking with Long Videos by Chain-of-Shot Reasoning, arXiv preprint arXiv: 2506.06097, 2025.

---

> ### Author Response · Authors · 2025-11-23
> **Response to Reviewer m7vg (Part 2 of 2)**
>
> **Q2:** While it uses fewer frames, the claim of "selecting key information" is not convincingly supported by the results.
>
> **A2:** Thank you for your valuable suggestion. Based on your feedback, we supplement both **quantitative** and **qualitative** experiments to more convincingly demonstrate that our method effectively selects key information.
>
> **1. Quantitative evaluation.** To quantitatively validate this capability, we conduct a **Needle-in-a-Haystack task** on Haystack-LVBench [2], and the results are shown in Table 5. This task evaluates the retrieval performance of task-relevant key frames using Precision, Recall, and F1. The results indicate that our method achieves higher retrieval performance than VideoAgent* and T* [2], while using either fewer or a comparable number of frames.
>
> This demonstrates that our approach does not simply reduce the number of processed frames—but more importantly, **it better selects key information**. This advantage stems from our multi-granularity perception mechanism, which captures key information more accurately than single-granularity captioning. Moreover, our verification mechanism mitigates caption hallucinations, further improving the accuracy of grounding key information.
>
> **Table 5. Experimental results on Haystack-LVBench. All retrieval metrics are reported in (%).**
> | Method                | Frames | Precision | Recall | F1  |
> |-----------------------|--------|-----------|--------|-----|
> | VideoAgent*           | 10.1   | 1.2       | 8.5    | 2.1 |
> | T*                    | 8      | 1.6       | 7.1    | 2.5 |
> | Ours w/o Verification | **7.8**| 2.3       | 8.9    | 3.7 |
> | Ours                  | 10.7   | **2.8**   | **10.6** | **4.4** |
>
> **2. Qualitative analysis.** To qualitatively illustrate the process of selecting key information, we provide an additional case study in Figure 4. This example shows how our method, through the collaboration between the LLM and the VLM, progressively locates key frames and verifies potential hallucinations, thereby identifying the truly critical information.
>
> [2] Re-thinking temporal search for long-form video understanding, CVPR 2025.
>
> **Q3:** The table formatting in the paper is poorly organized.
>
> **A3:** Thank you for your correction. We carefully revise all tables in the paper to ensure clearer formatting, appropriate placement, and improved readability.
>
> **Q4:** The paper should use \citep instead of \cite for citation.
>
> **A4:** Thank you for your correction. We have corrected all citations in the paper and consistently use \citep instead of \cite.

---

### Author Response · Authors · 2025-12-04
**Summary for the Area Chair**

**1. Summary of Reviewer Feedback**

We sincerely thank all reviewers for their constructive comments and recognition of our contributions, as summarized in the table below:

| Strengths                                      | m7vg | j8ZP | mkrX | ha9r |
|------------------------------------------------|------|------|------|------|
| Novel Perception-Verification                  |      |  ✅   |      |  ✅   |
| SOTA accuracy & efficiency                     |      |  ✅   |  ✅   |  ✅   |
| Thorough experiments                           |      |  ✅   |      |  ✅   |
| Training-free and effective                    |  ✅   |      |  ✅   |  ✅   |
| Well-written                                   |  ✅   |      |      |      |

**2. Reiterating Core Contributions**

As recognized by the reviewers, our core contributions to the community are:
1) **Hallucination Correction Mechanism**: We use the collaboration between LLMs and VLMs to cross-verify key information from different frame perspectives and granularities, preventing hallucinations in captions from interfering with LLM reasoning.
2) **Multi-granularity Perception**: We leverage the collaboration between LLMs and VLMs to capture key information at multiple granularities, thereby avoiding reliance on single-granularity retrieval and caption performance.
3) **SOTA accuracy & efficiency**: Extensive experimental results demonstrate that through our Perception-Verification loop, we achieve a SOTA accuracy-efficiency trade-off in a training-free manner.

**3. Key Clarifications**

In our responses, we provide detailed explanations and evidence to address specific concerns:

1) **Novelty (`j8ZP`, `mkrX`, `ha9r`)**: We clarify the essential differences between our method and existing approaches, including the **hallucination correction mechanism** and the **multi-granularity perception strategy**. In verification, through collaboration between LLMs and VLMs, we cross-verify key information from different frame perspectives and granularities, preventing the impact of caption hallucinations. In perception, we extract multi-granular key information using multiple strategies, reducing reliance on single-granularity retrieval and captioning performance in existing methods.

2) **LLM Agent Performance vs. Single Models and Its Necessity (`m7vg`)**: In fact, our performance **outperforms** single models like Qwen2.5-VL-7B and VideoChat-A1. We include additional results for Video-MME in Table 2. This is due to our method's ability to adaptively decide the scope and granularity of perception through the LLM Agent and correct hallucinations, avoiding the inherent limitations of single models with limited context lengths and hallucination issues.

3) **Grounding and Verification of Key Information (`m7vg`, `mkrX`)**: We demonstrate the grounding and verification of key information both **qualitatively** and **quantitatively**. **Quantitatively**, we add results for the **Needle-in-a-Haystack** task on Haystack-LVBench, as shown in Table 5. **Qualitatively**, we include a case of information selection and verification processes, as shown in Figure 4. These show that our method can effectively select key information, and the hallucination verification mechanism further aids in the localization of key information.

4) **Failure Case Analysis (`ha9r`)**: We add failure cases and analysis for both the perception stage and the verification stage, as illustrated in Figures 6 and 7. Perception failures are common when the question and its corresponding answer appear in different video segments, while verification failures are often due to extreme cases of consistent hallucinations across multiple frames.

**4. Revisions to the Manuscript**

We update the paper as suggested by the reviewers. **Changes are highlighted in the updated PDF**:

1) **Novelty (`j8ZP`, `mkrX`, `ha9r`)**: We emphasize the novelty of our method in the Introduction and Related Work sections.
2) **Additional Quantitative Experiments (`m7vg`, `mkrX`, `ha9r`)**: We add a comparison with proprietary and open-source MLLMs on Video-MME (Table 2). We also add results for the Needle-in-a-Haystack task on Haystack-LVBench (Table 5) and error bars with analysis (Figure 8).
3) **Additional Visualizations (`m7vg`, `mkrX`, `ha9r`)**: We add a case demonstrating the process of key information selection and verification (Figure 4) and a failure case in the verification stage (Figure 7).
4) **Formatting and Typographical Corrections (`m7vg`, `ha9r`)**: We reorganize the tables and correct typographical errors.

**5. Reviewer's Feedback on Our Response**

After carefully reviewing **all the other reviewers' comments and our responses**, Reviewer `j8ZP` decides to **maintain a positive rating**, for which we are sincerely grateful.

---

### Meta-Review · Area_Chair_8VnQ · 2026-01-09

**Summary:**

This submission proposes CogniGPT, an LLM agent framework for long video understanding leveraging a Perception-Reflection loop. Key reviewer concerns include limited conceptual novelty (recombining existing agentic, multi-granular retrieval, and verification ideas), initial gaps in validating key claims, and presentation issues. While the authors addressed some technical and formatting concerns, core novelty deficits remain unresolvable, leading to the rejection recommendation.

**Reviewer Concerns:**

- Addressed concerns: Table formatting, citation consistency, supplementary experiments (Video-MME comparisons, Needle-in-a-Haystack task), visualizations (key information selection/verification cases, failure analyses), and error bars for main datasets.
- Outstanding concerns: Core novelty (method builds on existing paradigms like VideoAgent/VideoTree/CoVe with incremental tweaks); insufficient demonstration of unique value beyond recombining established components.

**Reviewer Scores:**

- m7vg (initial: 2): Remains 2—performance clarifications do not resolve the necessity of the complex LLM agent.
- j8ZP (initial: 8): Remains 8—maintained positive rating post-rebuttal.
- mkrX (initial: 4): Remains 4—novelty concerns are not fully addressed.
- ha9r (initial: 4): Remains 4—added analyses do not mitigate core novelty gaps.

---

### Decision · Program_Chairs · 2026-01-26

Reject